# Microglia in Cultured Porcine Retina: Qualitative Immunohistochemical Analyses of Reactive Microglia in the Outer Retina

**DOI:** 10.3390/ijms24010871

**Published:** 2023-01-03

**Authors:** Kjell Johansson, Camilla Mohlin

**Affiliations:** 1Department of Science, Kristianstad University, 291 88 Kristianstad, Sweden; 2Department of Chemistry and Biomedicine, Linnaeus University, 391 82 Kalmar, Sweden

**Keywords:** retina, CD11b, Iba1, reactive microglia, synapse

## Abstract

A late stage of several retinal disorders is retinal detachment, a complication that results in rapid photoreceptor degeneration and synaptic damages. Experimental retinal detachment in vivo is an invasive and complicated method performed on anesthetized animals. As retinal detachment may result in visual impairment and blindness, research is of fundamental importance for understanding degenerative processes. Both morphological and ethical issues make the porcine retina a favorable organotypic model for studies of the degenerative processes that follow retinal detachment. In the cultured retina, photoreceptor degeneration and synaptic injuries develop rapidly and correlate with resident microglial cells’ transition into a reactive phenotype. In this immunohistochemical study, we have begun to analyze the transition of subsets of reactive microglia which are known to localize close to the outer plexiform layer (OPL) in degenerating in vivo and in vitro retina. Biomarkers for reactive microglia included P2Ry12, CD63 and CD68 and the general microglial markers were CD11b, Iba1 and isolectin B_4_ (IB_4_). The reactive microglia markers labeled microglia subpopulations, suggesting that protective or harmful reactive microglia may be present simultaneously in the injured retina. Our findings support the usage of porcine retina cultures for studies of photoreceptor injuries related to retinal detachment.

## 1. Introduction

In healthy adult brain and retinal tissues, microglia are resident immune cells with ramified morphologies whose thin processes continuously survey their microenvironment. The residential retinal microglial cells are localized to the inner retina and not found in the photoreceptor cell body layer. Thin microglial cell processes ramify into the synaptic plexiform layers, monitoring and regulating neuronal function and maintaining synaptic homeostasis [1,2]. The plexiform layers show complex synaptic arrangements, and the outer plexiform layer (OPL) is characterized by a precise synaptic circuitry between rod and cone photoreceptor axon terminals and interneurons [3,4]. Cone photoreceptors are responsible for daylight vision and trichromatic color detection in humans. These aspects of vision begin to form during neural processing at the synaptic level in the OPL [5]. Wang and colleagues [6] demonstrated by electroretinogram that microglia are required for synaptic maintenance in the OPL of healthy retina. In degenerative retinal diseases with visual impairment, the loss of photoreceptors and consequently, their synaptic connections in the OPL involve transition of the microglia into a reactive state [7,8,9,10]. 

In injured or diseased nervous tissues, reactive microglia may provide neuroprotection or promote neurodegeneration [11,12], clearly different processes that may depend on microglial cell heterogeneity. Human gene expression analyses suggest the presence of several microglial populations and that the heterogeneity emerges during neurodegenerative disorders [13]. Recent electron microscopic and immunoelectron microscopic studies on various brain areas show that microglia engulf parts of the synaptic elements resulting in the remodeling of neural circuitry in the developing and mature nervous system [14]. In a series of studies [15,16], a dark microglia phenotype has been thoroughly described in the context of synaptic damage during neurodegenerative disorders and injuries.

Analyses of microglia during retinal diseases and injuries have mainly been focused on neuroinflammation, as well as migration of reactive microglia into the ONL and phagocytosis of apoptotic photoreceptors and outer segments [9,10,17,18,19]. Apart from their contribution to photoreceptor cell death, these studies show that reactive microglia are juxtaposed to and project their processes into the OPL. Recent studies [6,20] show that microglia promote photoreceptor survival, indicating that microglial function may differ depending on the localization in the retinal parenchyma (e.g., OPL vs. subretinal space). Whether or not the microglial heterogeneity depends on animal models and/or environmentally induced microglial cell plasticity has yet to be determined.

Several the organotypic retinal studies are focused on the establishment of an in vitro model for severe diseases, such as age-related macular degeneration (AMD), diabetic retinopathy (DR), retinal detachment (RD) and glaucoma [21,22,23]. Such studies on porcine retinas are growing because of the histological and ethical advantages. The porcine retina has a cone-enriched area centralis resembling the human macula, and ocular material is considered waste material in food industries [7]. Coculture paradigms in which retinal explants are cocultured with supporting cells (e.g., neural progenitor cells or ARPE-19 cells) show beneficial effects on photoreceptor survival and synaptic integrity and also complement activation, as well as neural and glial remodeling [7,24,25,26,27].

This study identified reactive microglia using established markers including CD11b, CD68, Iba1 and P2Ry12 [28]. In particular, the distribution of microglial cell bodies and their projection of processes close to or within the OPL was studied by double staining using mouse antibodies raised against the horizontal cell marker calbindin, which also may label subpopulations of amacrine and bipolar cells [29] or the photoreceptor synaptic marker PSD95 [30]. Microglia were in some cases also identified using isolectin B_4_ [31] in combination with the antibodies mentioned above. Antibodies raised against CD63 [32] were used to examine putative intercellular communication by microglial-derived extracellular vesicles under degenerative conditions. 

## 2. Results

### 2.1. Overview of Microglia Associated with the OPL in Healthy and Degenerating Retina

Immunohistochemical identification of microglia was done by using antibodies raised against CD11b and Iba1 as general microglia markers in healthy and degenerating retina. Usage of these antibodies resulted in similar labeling patterns: in healthy adult retina, immunolabeled microglial cell bodies localized to the inner retina and projected long, thin processes through the parenchyma that finally terminated close to photoreceptor terminals at the inner aspect of the OPL (Figure 1A). During degeneration in vitro, the microglia altered localization and their processes became thicker. The localization of immunolabeled cell bodies and the entry of their processes into the OPL was evident after 24 h of culture. After five DIV, immunolabeled microglial cell bodies localized close to the OPL and emanated processes into and along the neuropil. Immunolabeled microglia were also evident within the ONL and among photoreceptor outer segments (Figure 1B,C). 

Migration of microglia was also correlated with morphological changes. In normal retina, the Iba1-immunolabeled microglia showed a ramified appearance with thin processes into the surrounding parenchyma (Figure 1A). When associated with the OPL in five DIV retinas, the different microglial biomarkers revealed varying morphological shapes of the labeled cell bodies from round to elongated (Figure 1B,C). In general, the labeled cell processes were rather thick and evident close to photoreceptor synaptic terminals (Figure 1B). 

### 2.2. Reactive Microglia in Degenerating Retina

#### 2.2.1. IB_4_

Isolectin B_4_ (IB_4_) is a potent microglial cell marker in degenerating but not in normal adult porcine retina. With time in vitro and increasing degenerating parenchyma, microglial alterations were evident with respect to their cellular morphology and localization close to the OPL. In the five DIV specimens, IB_4_-labeled microglia localized to the INL, OPL, ONL and among degenerating photoreceptor outer segments (Figure 1C). At this time, two morphologically different IB_4_-labeled microglia were also evident as described above: large microglia with a rounded cell body and small microglia with an elongated cell body. Residual retinal vessels also displayed IB_4_ labeling at this time point (Figure 1C).

#### 2.2.2. P2Ry12

Immunoreactivity for the purinergic receptor P2Ry12 in the brain has been identified on microglial cell processes [28]. P2Ry12 senses ATP in injured nervous tissue and mediates the extension of processes early in reactive microglia [33,34,35,36]. The P2Ry12 antibody was raised in rabbits and stained well only with the usage of detergents, Colabeling was therefore performed with the neural markers calbindin and PSD95. Dotted P2ry12-immunoreactivity was mainly observed in the inner 2/3 of labeled Müller cells in normal retina (Figure 2A,B). After five DIV, weak P2ry12-immunoreactivity persisted in the Müller cells and was now also evident in the OPL. The P2ry12-immunoreactivity in the OPL of cultured specimens was evident as dots or small profiles throughout the synaptic layer (Figure 2C,D). This labeling pattern was partly interpreted as labeled microglial processes: colabeling analyzes showed weak P2ry12-immunoreactivity in some presumptive microglial cell bodies (arrow in Figure 2 H). P2ry12-immunoreactivity was separated horizontal cell processes (Figure 2E,F) but observed juxtaposed to degenerating photoreceptor synaptic terminals (Figure 2G). P2ry12-immunoreactivity could occasionally be observed in presumptive cone photoreceptor axon terminals (arrows in Figure 2E).

#### 2.2.3. CD63

CD63 is commonly used as a marker of late endosomes and multivesicular bodies and may be enriched on extracellular vesicles in various tissues, including adult retina [32]. Similar with P2ry12, weak CD63-immunoreactivity was only observed in Müller cells in normal adult retinas (Figure 3A). The immunoreactivity showed an uneven distribution and was evident in the inner 2/3 of the labeled Müller cells: this labeling pattern persisted in the five DIV retinas. With time in vitro, CD63 immunoreactivity begun to appear in single IB_4_-labeled reactive microglia localized close to the inner aspect of the OPL. Fluorescence (Figure 3B,D) and confocal microscopy (Figure 3C) revealed a perinuclear distribution pattern of CD63 immunoreactivity in IB_4_-labeled microglia located close to the OPL (Figure 3B,D). At five DIV, vesicular CD63 immunoreactivity was also observed in the presumptive reactive microglia in the ONL (arrows in Figure 3E). Perinuclear and vesicular distribution of the immunoreactivity may represent CD63 labeling in the Golgi network and in late endosomes, respectively. 

#### 2.2.4. CD68

In line with previous data [28], antibodies raised against the lysosomal marker CD68 were used to identify reactive microglia. No CD68 immunoreactivity could be observed in healthy retina. In the degenerating 5DIV retina, single CD11b- and Iba1-immunoreactive microglia in the ONL and close to the OPL displayed CD68 immunoreactivity (Figure 4A,B). The CD68-immunoreactivity was mainly localized to the cell body and in subcellular structures most likely representing lysosomes (Figure 4C). Despite the weak labeling of small structures like lysosomes, the presence of CD68 immunoreactive structures could be demonstrated in CD11b and Iba1 immunoreactive processes aligned close to the OPL by confocal microscopy (small arrows in Figure 4D–F). Weak CD68 and Iba1 immunoreactivities were also evident in small microglial cell bodies (large arrows in Figure 4E,F) localized close to the OPL.

## 3. Discussion

### 3.1. General Considerations

The present study focuses on light-illuminated five DIV retinas, a time-point with detectable photoreceptor degeneration, as well as the migration and altered morphology of reactive microglia. At 5 DIV, CD11 or Iba1 immunoreactive microglia had migrated not only towards and arborized within the OPL but were also found in the ONL and among the outer segments [7]. The appearance of reactive microglia temporally coincided with the in vitro-induced photoreceptor synaptic damage in the OPL, a feature immunohistochemically detectable after 48 h [27,37]. 

At five DIV, CD63 and P2Ry12 immunoreactivities were mainly observed in reactive microglial cell bodies and their processes in the OPL, respectively. CD68 immunoreactivity was at this time point mainly observed in CD11b and Iba1 immunoreactive microglia in the ONL. The P2Ry12 immunoreactive profiles in the OPL were primarily observed in the immediate vicinity of degenerating PSD95 immunoreactive photoreceptor terminals but separated from horizontal cell processes. P2Ry12 immunoreactive microglia cell bodies were scarce at five DIV. Also, the densities of CD63 and CD 68 immunoreactive microglial cell bodies were very low and only observed in single colabeled microglia. Thus, the few labeled microglia were reactive, but they displayed different immunoreactivities that may implicate different phenotypes. 

### 3.2. Retinal Detachment in the Cultured Retina

The separation of the retina from the underlying pigment epithelium, or retinal detachment, is a complication in several retinal disorders that result in vision impairment [4,38]. Retinal detachment in vivo represents an experimental approach for histological analyses of photoreceptor degeneration and synaptic damages. Similarly, the culturing of retina results in an acute retinal detachment, resulting in degenerative processes that mimic the ones described in retinal detachment in vivo [39]. In vivo studies show that rod photoreceptor synaptic damage occurs as early as two hours post-detachment [36] and that photoreceptor death begin within 12 h of the injury [20]. Both rod and cone photoreceptors show synaptic damage after detachment, rods by retracting spherules and cones by altered pedicle morphology and the loss of intracellular ribbons [40,41,42]. Alteration of ribbon synapses develops following protein accumulation in photoreceptors in vivo [43]. The impact of reactive microglia and synaptic damages in vivo is not well-known but increases in microglia in the OPL in the degenerating retinoschisis mouse retina [44], as well as following light-induced retinal injuries [45], have been reported. In the study by Wang and colleagues [6], it was shown that depletion of microglia resulted in degeneration of photoreceptor synapses and impaired light response. Depleting microglia is also known to inhibit the death of photoreceptors by regulating neuroinflammatory processes [20]. However, another study reported that microglia rather contributed to the loss of photoreceptors [46].

Most of the in vitro studies, mostly on rodent retinas but also on the porcine retina [24,25,26,27], have focused on a variety of rescue paradigms and technical improvements to preserve photoreceptor survival [22,47]. In the porcine retina, neuroprotection in vitro demonstrated an increased density of microglia at the OPL [7], but the significance is not known. Previous studies have shown the presence of rapid synaptic damage in cultured porcine retinas [6,48,49,50] and after eight hours/day-light exposure in vitro [37]. Synaptic damages appear to develop prior to the detection of photoreceptor death and result in cone pedicle flattening and retraction of rod axons. A recent study indicated that the coculture of porcine retina and human neural progenitor cells supported synapse structure in the OPL [25]. Townes-Anderson and colleagues [41] concluded that synaptic damage must be reduced to preserve vision during photoreceptor degeneration.

### 3.3. Microglia Heterogeneity

As mentioned above, there exist contradictory results concerning the function of microglia in retinal detachment: beneficial or detrimental. Such results may depend on the species studied but also on presence of different microglial cell subpopulations. A recent study [51] demonstrated that microglia functional heterogeneity depends on location in the retinal parenchyma. If this transition from one function to another has a temporal aspect, it seems reasonable to assume that both protective and harmful microglia are present simultaneously in the injured parenchyma. Apoptotic cells have recently been shown to affect microglia heterogeneity, and several functional microglial types coexist in the developing retina [52].

Reactive microglia also express P2Ry12 immunoreactivity, a receptor known to be responsive to ADP and ATP from injured neurons [53]. One possibility is that the P2Ry12 immunoreactivity described in OPL identifies protruding microglial processes [36] that have extended into the OPL. It may be possible that the OPL-associated microglia interact with the injured capillaries localized at the inner aspect of the OPL (see [54] for capillary-associated microglia) and protect injured capillaries and/or neural structures [38], such as cone synapses. In healthy adult retina, CD63 labels extracellular vesicles (EVs), which carry diverse cargo for intercellular communication within the parenchyma [32]. In the context of retinal degeneration, the characterization of EVs is scarce [55], but a recent study suggested that miRNA in EVs is important for the maintenance of normal retinal homeostasis [56]. Microglia-derived EVs have also been suggested to contribute to the progression of retinal pathology [55]. If the CD63 immunoreactive microglia described in the current study release EVs can yet only be speculated upon, but small CD11b immunoreactive profiles were evident using confocal microscopy. A small population of CD68 immunoreactive microglia localized close to OPL and in the ONL in five DIV porcine retinas. CD68 is a lysosomal marker and CD68 immunoreactive microglia and phagocytic function have been shown in cell culture [57]. This indicates that the reactive microglia in the ONL and OPL may be detrimental at five DIV. 

## 4. Material and Methods

### 4.1. Animals and Culture Paradigm

Adult porcine retinas (*n* = 4–6) were processed and cultured (under light illumination 8 h/day) for five days in vitro (5 DIV) as described previously [7]. For light and confocal microscopy, cultured retinas were immersed in 4% paraformaldehyde (Sigma-Aldrich, St Louis, MI, USA) in 0.1 M Sorensen’s phosphate buffer (Sigma-Aldrich) for two hours at 4 °C.

### 4.2. Fluorescence and Confocal Microscopy

Fixed retinas were washed, cryoprotected in 20% sucrose and sectioned at 10–12 µm. Antigen retrieval was performed by heating the sections in 0.01 M sodium citrate pH 6.0 for 3 × 15 s in a microwave oven at 450 W, followed by washes with phosphate buffer (PBS; Gibco, Paisley, UK) for 30 min at 4 °C. In double-labeling experiments, the sections were incubated with an appropriate mixture of primary antibodies (see Table 1) and diluted in PBS containing 0.25% Triton-X (Sigma-Aldrich) overnight at 4 °C. After washes, sections were incubated with VectaFluor^TM^ double-labeling kit RTU containing DyLight^®^488 anti-rabbit IgG and DyLight^®^594 anti-mouse IgG (Vector Laboratories, Burlingame, CA, USA) for 30 min at room temperature. The sections were treated according to instructions given by the manufacturer and eventually mounted in Vectashield containing DAPI (Vector Laboratories).

Fluorescence images were viewed and captured using an Olympus BX60 (Olympus-Europe, Hamburg, Germany) with appropriate filter settings, and images were captured 60× using a digital acquisition system (Olympus DP74 camera). Confocal images were captured and processed using a Leica SP8 (Leica Microsystems, Wetzlar, Germany) confocal microscope with appropriate filter settings for 488 and 594 nm, as well as DAPI using LAS X software (Leica Microsystems, Wetzlar, Germany). Adobe Photoshop (Adobe CC 2017, Adobe Systems, San Francisco, CA, USA) was used to adjust contrast and brightness. 

### 4.3. Lectin Histochemistry and Immunohistochemistry

Further identification of reactive microglia was performed on some immunohistochemically labeled sections by using the common microglia marker isolectin B_4_ (IB_4_) [31], conjugated with fluorescein isothiocyanate (FITC) (1:1000; Vector Laboratories). The usage of isolectin in double-labeling experiments depends on a limited number of antibodies that detect microglia in porcine retina. The primary antibody was detected using DyLight^®^594 anti-mouse IgG (Vector Laboratories), whereafter the lectin was applied for 30 min at room temperature in darkness. The sections were rinsed and counterstained as previously.

Conventional detergents are usually used to improve antibody penetration and in the detection/labeling of intracellular proteins/structures. The usage of detergents in combination with lectin histochemistry resulted in impaired labeling and was avoided; colabeling approaches using IB_4_ and antibodies were used to identify microglial cell bodies.

## 5. Conclusions

Even though the current focus is on the immunohistochemical identification of different reactive microglia at the OPL, we may conclude that a few microglia express biomarkers indicating different functional phenotypes. One interpretation is that the resident microglia develop into a heterogeneous population that localize in different compartments of the degenerating in vitro retina. The porcine retina is a suitable model to study early degenerative processes that developed following retinal detachment. By using pharmacological agents, growth factors or coculture with supporting cells, the event(s) that trigger the transition of resident microglia into a reactive phenotype can be further investigated in vitro.

## Figures and Tables

**Figure 1 ijms-24-00871-f001:**
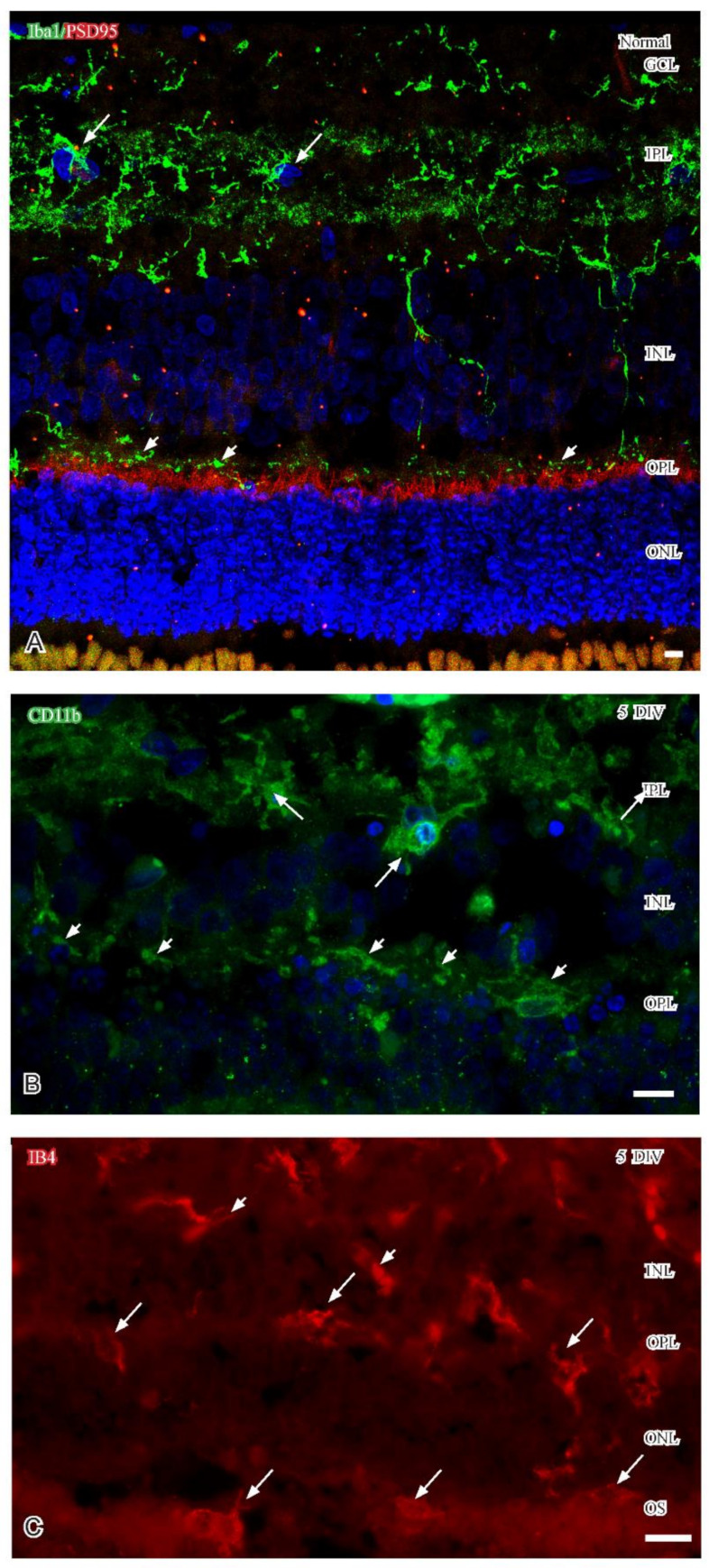
Confocal stacks (**A**,**B**) and fluorescence micrographs (**C**) of normal and 5 DIV retinas. (**A**) In normal healthy retina, Iba1 immunoreactive microglia (green: large arrows) localize to the IPL and project long, thin processes that finally arborize along the inner aspect (small arrows) of the PSD immunoreactive photoreceptor terminals (red) in the OPL. At 5 DIV (**C**), retinas CD11b immunoreactive microglia (green) are evident at the IPL (large arrows), close to photoreceptor terminals in the OPL (small arrows). Thick CD11b immunoreactive processes localize close to or within the OPL (small arrows). (**C**) 5 DIV retina with IB_4_ labeled microglial cell bodies (large arrows) localized in the outer retina. Note also labeled degenerating capillaries (small arrows). DAPI counterstaining in (**A**,**B**). Abbreviations. GCL ganglion cell layer; INL inner nuclear layer; IPL inner plexiform layer; ONL outer nuclear layer; OPL outer plexiform layer; OS outer segments. Scale (**A**–**C**) 10 µm.

**Figure 2 ijms-24-00871-f002:**
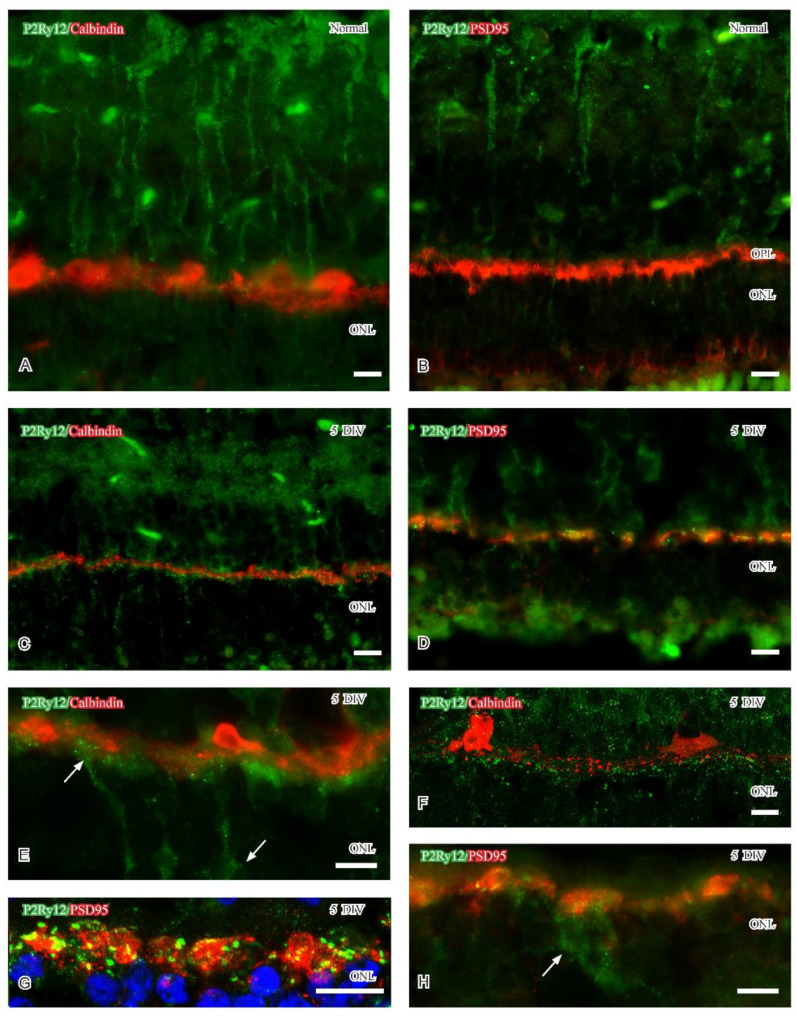
Confocal stacks (**F**,**G**) and fluorescence micrographs (**A**–**E**,**H**) showing P2Ry12 immunoreactivity (green) in normal and 5 DIV retina. Calbindin or PSD95 immunoreactivity (red) indicate the OPL. (**A**,**B**) P2Ry12 immunoreactivity is mainly evident in the inner parts of Müller cells. (**C**,**E**,**F**) Colabeling with calbindin shows that P2Ry12 immunoreactivity distributes close to horizontal cell processes at 5 DIV. (**E**) Structures resembling cone photoreceptor terminals occasionally expressed P2Ry12 immunoreactivity. (**D**,**G**) Juxtaposed P2Ry12 and PSD95 immunoreactivities in the OPL (yellowish hue in (**D**)). (**H**) P2Ry12 immunoreactive microglial cells (arrow) and processes in the PSD95 labeled OPL. Abbreviations as in Figure 1. Scale (**A**–**H**) 10 μm.

**Figure 3 ijms-24-00871-f003:**
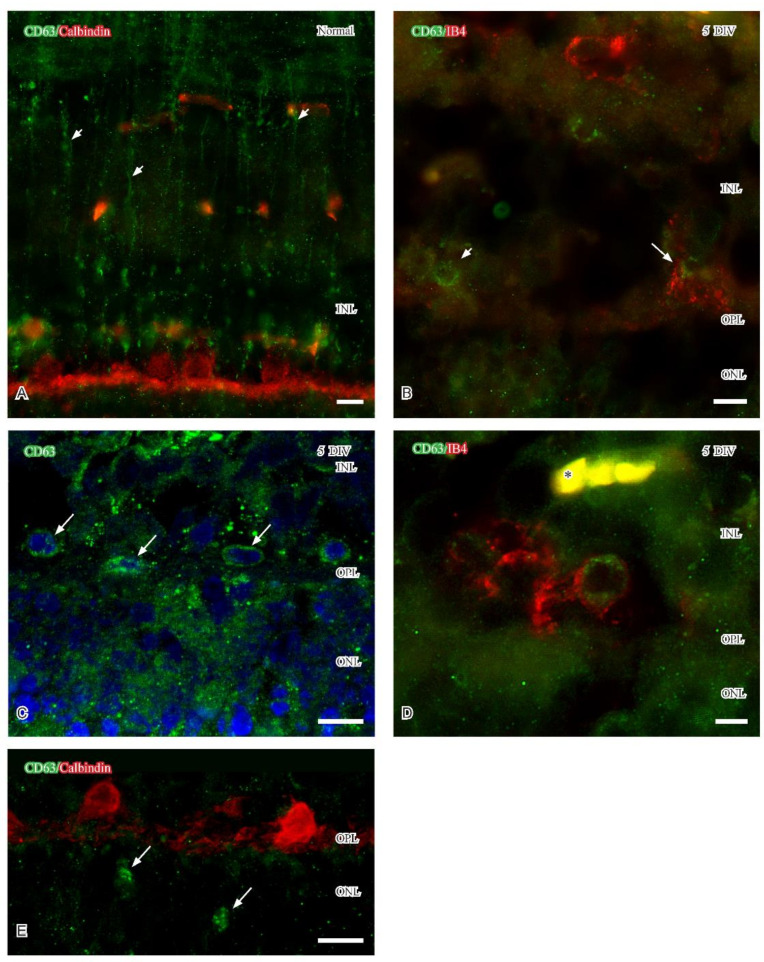
Confocal stacks (**C**) and fluorescence micrographs (**A**,**B**,**D**) showing CD63 immunoreactivity (green) in normal and 5 DIV retina. Calbindin immunoreactivity and IB_4_ labeling (red) indicate horizontal and microglial cells, respectively. (**A**) Dotted CD63 immunoreactivity is evident in Müller cells in healthy retina. (**B**) IB_4_-labeled microglia close to the OPL with perinuclear CD63 immunoreactivity (large arrow). Note the presence of CD immunoreactive cell without IB_4_ labeling (small arrow). (**C**,**D**) High magnification of the OPL showing perinuclear CD63 immunoreactivity (arrows in **C**) and in IB_4_-labeled microglia (**D**). Note also labeled capillary (asterisk in (**D**)). (**E**) CD63 vesicular immunoreactivity in presumptive microglia (arrows) in ONL. Abbreviations as in Figure 1. Scale (**A**–**E**) 10 μm.

**Figure 4 ijms-24-00871-f004:**
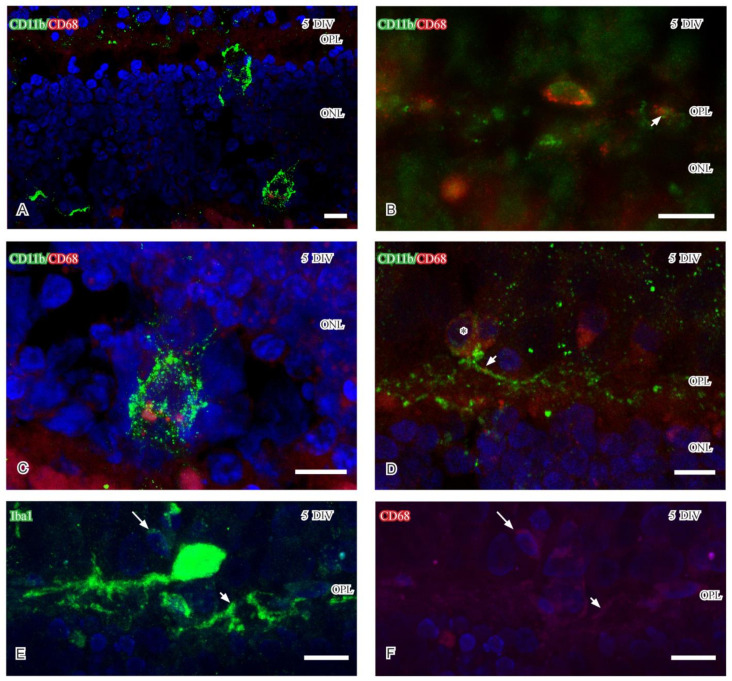
Confocal stacks (**A**,**C**) and fluorescence micrographs (**B**,**D**) showing CD68 immunoreactivity (red) and CD11b immunoreactive microglia (green). Confocal stacks (**E**,**F**) showing CD68 immunoreactivity (red) in Iba1 immunoreactive microglia (green). (**A**,**C**) Vesicular CD 68 immunoreactivity in low and high magnification in the ONL of 5 DIV retina. (**B**,**D**) Perinuclear CD68 in a CD11b immunoreactive microglia close to the OPL. (**D**) CD68 immunoreactivity in microglial cell body (asterisk) and process (small arrows in **B**,**D**). (**E**,**F**) CD68 immunoreactivity in large strongly Iba1 immunoreactive microglial cell and processes (small arrows in (**E**,**F**)). Note also small microglia cell body (large arrows in **E**,**F**) displaying weak CD68 and Iba1 immunoreactivities. Abbreviations as in Figure 1. Scale (**A**–**F**) 10 µm.

**Table 1 ijms-24-00871-t001:** Antibodies and lectins.

Antibodies Used, Host Animal (Rb = Rabbit, Mo = Mouse), Dilution, Company and Reference	
Microglia markers.	
CD11b Rb 1:2000; Abcam, Cambridge, UK (general marker)	[28]
CD63 Rb 1:100, Abcam, Cambridge, UK (reactive microglia; EV marker)	[32]
CD68 Mo 1:100 Santa Cruz Biotechnology, Dallas, TX, USA (reactive microglia; lysosomal marker)	[28]
Iba1 Rb 1:1000; Proteintech Group, Chicago, IL, USA (general marker)	[28]
P2yR12 Rb 1:100 Novus Biological, Abingdon, UK (reactive microglia)	[28]
Neural markers	
Calbindin Mo 1:2000, Santa Cruz Biotechnology, Dallas, TX, USA (horizontal cells)	[29]
PSD95 Mo 1:1000; Millipore, Burlington, CA, USA (photoreceptor synapse)	[30]
Lectins	
Isolectin B4 FITC- or Texas red-conjugated lectin 1:1000, Vector Laboratories, Burlingame, CA, USA (general microglia marker but only in the degenerating porcine retina).	[31]

## Data Availability

Data sharing is not applicable to this article.

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
