# Peer review of "Microglia in Cultured Porcine Retina: Qualitative Immunohistochemical Analyses of Reactive Microglia in the Outer Retina"

_ijms, 2023, doi:10.3390/ijms24010871_

Round 1

Reviewer 1 Report

Thank you for submitting your manuscript. Overall the paper is well structured, straightforward and easy to understand. The results are quite clear and support the conclusions.

The images could be better as some of them are a bit pixelated and could be of better quality. 

Also the methods could be extended a bit and be more descriptive for instance, in terms of the filters used.

Minor typing errors to be corrected.

Author Response

Reviewer 1

Filter settings for confocal microscopy has been added.

Material and methods

Para 2.2, line 97-98: for 488 and 594 nm as well as DAPI using

Figures

I´ve tried to find out the pixelated pictures and suspect Fig 4 E and F. In the revised version, the micrographs smaller and have hopefully gained more pixels.

Typing errors have been corrected

Reviewer 2 Report

Introduction:

Line from 60-62: Can you add some more references? you talk about several studies but you only have one reference.

Line 72: calbindin is not a horizontal cell marker exclusively. It would be nice if you specified that mark different subpopulations of cells in the retina including the horizontal cells (you can reference Guduric-Fuchs et al., 2009: https://pubmed.ncbi.nlm.nih.gov/19784390/).

Material and Methods:

I miss the reference number on the antibody list.

Results:

Figures: it would be nice if you provide the DAPI channel to distinguish better the different layers.

Author Response

Intro, line 60-62: 2 additional references concerning in vitro models and retinal degeneration have been added. All 3 references have recently been published 2020-2021, and present new retinal in vitro data.

Intro line 72: following sentence has been added …….which also may label subpopulations of amacrine and bipolar cells [29]

References have been added to the antibody list

We have only used dapi filter with confocal microscopy, and dapi information is unfortunately not available for pictures taken by fluorescence microscopy.
